# How Prostate Cancer Cells Use Strategy Instead of Brute Force to Achieve Metastasis

**DOI:** 10.3390/cancers11121928

**Published:** 2019-12-03

**Authors:** Darron Tharp, Srinivas Nandana

**Affiliations:** Department of Cell Biology and Biochemistry, Texas Tech University Health Sciences Center, Lubbock, TX 79430, USA; Darron.Tharp@ttu.edu

**Keywords:** prostate cancer, metastasis, strategies for metastasis, tropism, immune evasion

## Abstract

Akin to many other cancers, metastasis is the predominant cause of lethality in prostate cancer (PCa). Research in the past decade or so has revealed that although metastatic manifestation is a multi-step and complex process that is orchestrated by distinct cellular and molecular mechanisms, the process in itself is an extremely inefficient one. It is now becoming increasingly evident that PCa cells employ a plethora of strategies to make the most of this inefficient process. These strategies include priming the metastatic sites ahead of colonization, devising ways to metastasize to specific organs, outsmarting the host defense surveillance, lying in a dormant state at the metastatic site for prolonged periods, and widespread reprogramming of the gene expression to suit their needs. Based on established, recent, and evolving lines of research, this review is an attempt to understand PCa metastasis from the perspective of military combat, wherein strategic maneuvering instead of brute force often plays a decisive role in the outcome.

## 1. Introduction

Similar to many other cancers, metastasis is the predominant cause of morbidity and mortality in advanced prostate cancer (PCa). Although patient survival rates have improved over the past few years for localized disease, patients with metastatic disease do not share these improvements. This isunderscored by the fact that eight of the 12 cancers followed for a decade (2005–2015) showed a decrease in the 5-year survival rates of patients diagnosed with metastatic disease, and PCa was among these cancers [1].

Unfortunately, we know much less about the biology of metastasis than we know about primary tumors. However, this paradigm is beginning to shift as recent research has been particularly insightful in shaping our understanding of the broad principles that determine metastatic manifestation, as well as the specific molecular and cellular players that drive metastasis. Research in the past decade and a half has revealed that metastasis is a complex and multi-step coordinated process, and that each step is dictated by distinct molecular and cellular mechanisms. The metastatic cascade encompasses: (a) growth and invasion of the primary tumor cells that breach the surrounding basement membrane; (b) intravasation into the blood stream; (c) survival in circulation; (d) extravasation of the tumor cells at distant sites, and finally; (e) metastatic colonization. We also have come to comprehend that the successful execution of these individual steps, culminating in the establishment of metastatic disease, is mediated in a large part by heterotypic interactions between cancer cells and the tumor microenvironment, which acts as a key accomplice.

Although for the overwhelming majority of patients, metastatic disease is associated with a very high burden of lethality, it is perplexing that the metastatic process per se is an extremely inefficient one—only about 0.02% of cancer cells that escape the primary tumor are successful in establishing overt metastases [2,3,4,5]. Through a combination of factors including immune surveillance, hemodynamic shear stress, and prospective niche incompatibility, the majority of cancer cells that escape the primary site are rendered ineffective [3]. It is now understood that the rate-limiting step of successful metastasis lies largely in the ability of the cancer cells to adapt and grow in the foreign microenvironment, wherein the local microenvironmental milieu can be vastly different from that of the primary tumor. Despite these overwhelming odds, an evolving body of research is highlighting the fact that PCa cells win the metastasis war by employing strategy, and that these strategies predominantly revolve around engaging the microenvironment to benefit the cancer cells, a process that ultimately tips the odds to favor the establishment of overt metastasis.

This review is an attempt to provide a perspective in comprehending the scaffold of the strategies utilized by disseminated PCa cells by drawing parallels between the metastatic cascade and military combat, and thereby appreciate the uncanny resemblance between the two (Figure 1). A thorough understanding of the framework of these strategies will translate into novel therapeutic modalities that are specifically tailored to curtail the cross-talk between PCa cells and the metastatic microenvironment, thereby delaying or halting the metastatic progression.

The strategies (Figure 1) used by PCa cells can be summed up and discussed as follows:

### 1.1. Metastatic Tropism (Global Positioning System)

We now know that cancer cells do not metastasize in a random manner, and largely follow organo-tropism mainly dictated by specific molecular and cellular interactions with the metastatic microenvironment—similar to troops using the Global Positioning System (GPS) in modern military combat (Figure 1). The beginnings of our understanding of the principles of metastatic dissemination dates back more than a century ago when Stephen Paget made the seminal observation that although tumor cells are broadly disseminated during metastatic progression, detectable or overt metastases only develop at particular sites or “soils” that are preferentially adapted for the survival and proliferation of tumor cells or “seeds.” This observation formed the basis of Paget’s “seed and soil” hypothesis, which propounds that the overt pattern of clinical metastases cannot be explained by the anatomical layout of the vasculature, and that the missing crucial factor is the adaptability of the tumor cells in the particular foreign environments [6]. Subsequent findings from several research groups on metastatic colonization of diverse cancers have been consistent with Paget’s hypothesis that specific organ microenvironments are more or less adaptable for the metastatic colonization of specific types of tumor cells. 

The mechanisms by which PCa cells achieve this tropism are varied and include: a) secretion of cytokines and growth factors by the primary tumor, which prime the specific organ microenvironments prior to arrival of the tumor cells, also known as the “pre-metastatic niche”; b) mechanisms through which tumor cells adapt and grow in an advantageous manner in the foreign microenvironments through production of factors e.g., prostate and breast cancer cells are known to secrete growth factors that mediates a vicious cycle of bone remodeling, leading to clinical complications of fractures and pain, and; c) mechanisms that facilitate tumor cell homing to specific organ sites e.g., PCa cells are known to mimic or parasitize homing mechanisms utilized by hematopoietic stem cells (HSCs) in colonizing the bone. 

A vital mechanism by which PCa cells home to the bone is mediated by Chemokine (C-X-C motif) Receptor type 4 (CXCR4)/Chemokine (C-X-C motif) Ligand 12 (CXCL12) signaling, and multiple investigations have documented the seminal role of CXCR4/CXCL12 chemokine axis in PCa homing to the bone [7,8,9,10]. Binding of the CXCL12, also called stromal cell-derived factor-1 (SDF-1), to the G-protein coupled receptor CXCR4 mediates a plethora of functions that drive tumor growth and metastasis through chemotaxis, cell survival, proliferation, and angiogenesis. CXCR4 overexpression in PCa is associated with an aggressive phenotype and poor survival rates [11], CXCR4 expression correlates with increasing tumor aggressiveness, and PCa metastases display elevated CXCR4 levels compared with primary tumors [9]. The interaction of CXCR4/CXCL12 and the resultant downstream signaling pathways orchestrate multiple responses in PCa cells including the survival-associated MEK/ERK signaling and activation of NF-kappa B [12,13]; and increased interaction of PCa cells with the endothelial cells or the stromal collagen fibronectin and laminins that occurs by the upregulation of integrins [13,14]. Mouse model studies have revealed that an antibody against CXCR4 reduced bone metastases, and a blocking peptide against CXCR4 hindered intra-osseous PCa growth [8]. In addition, the CXCR4/CXCL12 axis mediates angiogenesis in PCa, as evidenced by the observation that CXCR4 over-expressing subcutaneous xenografts of PC3 cells, had increased vascularization and a more invasive phenotype [15].

Another intriguing facet that involves CXCR4/CXCL12 signaling and PCa bone metastasis pertains to the homing of HSCs to the bone. CXCL12 is constitutively and highly expressed by several cells of the bone niche including osteoblasts, endothelial cells, and other bone stromal cells, wherein its interaction with CXCR4 mediates HSC homing to the bone marrow [16,17]. In fact, disruption of CXCR4/CXCL12 mediated signaling is a vital pre-requisite for HSC mobilization from the bone marrow into the blood, as demonstrated by the success of CXCR4 inhibitor plerixafor as a tool for the rapid mobilization of HSCs into human blood [18]. Further, an elegant study found that PCa cells directly compete with HSCs to occupy the endosteal niche, indicating that homing mechanisms can direct bone niche occupation in PCa [19]. This report also raised the possibility that agents which result in the mobilization of HSCs from the niche may also cause mobilization of PCa cells, and indeed the study found that mice when treated with plerixafor or G-CSF (granulocyte colony-stimulating factor), that are both known to mobilize HSCs, resulted in the exodus of PCa cells from the bone marrow [19]. The study raised a seminal clinical implication for the treatment of PCa bone metastasis, since signals that mediate cancer cell niche-dependent behavior could provide therapeutic niche disruption strategies that would render PCa bone metastasis more susceptible to chemotherapeutic intervention.

### 1.2. Pre-Metastatic Niche (Sending a Reconnaissance Team Ahead of the Troops)

There is growing evidence that suggests that the metastatic progression of the primary tumors is initiated much earlier than previously thought—thereby preparing the metastatic site for subsequent colonization by the cancer cells. In the war analogy, this is akin to sending a reconnaissance team ahead of the troops (Figure 1). The pre-metastatic niche can be described as a tissue microenvironment or “soil” that is undergoing a set of molecular and cellular changes to make it more supportive and receptive for metastatic colonization by the cancer cells or “seed.” Research on the pre-metastatic niche has been pioneered by Dr. Lyden and his colleagues [20], and the importance of this niche in driving metastasis has been increasingly appreciated in recent years. The pre-metastatic niche within the future metastatic sites can be initiated and established through: (a) primary tumor-derived components; (b) tumor-mobilized bone-marrow-derived cells (BMDCs), and; (c) the local stromal microenvironment of the host—all working hand-in-hand to render the distant organs hospitable for the colonization of disseminated tumor cells. The primary-tumor derived molecular components include tumor-derived secreted factors (TDSFs), extracellular vesicles (EVs), and other molecular components that include cytokines, chemokines, and inflammatory factors produced by the tumor cells.

EVs serve as an important means of communication between cancer cells and its surrounding cells. The cargo and the membrane lipids of EVs contain a variety of components including cytoplasmic proteins, nucleic acids, and lipids—a composition that is reflective of the parent or producer cell but is also made up of a unique mixture of proteins and genetic material [21,22,23]. EVs can function by directly stimulating the target cells through interactions with ligands expressed on the surface, transferring membrane receptors between cells, and through the horizontal transfer of proteins and genetic information [21,24,25,26,27,28,29,30]. A growing body of evidence has implicated the cargo in the exosomes—the most widely studied class of EVs—in driving varied facets of PCa progression and metastasis. For example, exosomes from the prostate tumor microenvironment can promote EMT through the repression of tumor suppressor genes such as Ras suppressor 1 and stromal antigen 2 [31], and tumor growth factor beta (TGF-β) from PCa-derived exosomes has been demonstrated to play a role in the conversion of bone marrow mesenchymal cells into high VEGF and HGF secreting myofibroblasts [32].

In addition to the primary tumor microenvironment, there is growing evidence that EVs can be released in several body fluids, as has been reported in PCa [33], and therefore EVs are thought to play a role in the metastatic spread of PCa. In line with this hypothesis, miRNAs from EVs have been shown to correlate with PCa metastasis and several reports have suggested that these miRNAs could be utilized as potential biomarkers and therapeutic targets for PCa [34]. miRNA-141 and miRNA-375 have been reported to be involved in metastatic PCa and were shown to be increased in plasma or serum-derived EVs from metastatic PCa patients [35]. Moreover, expression of miRNA-141 was found to correlate with tumor grade and metastasis [36,37,38]. Other reported exosomal-origin miRNAs associated with diagnosis or prognosis of PCa include Let-7c [39], miRNA-1290, and miRNA-375 [40]. Furthermore, miRNA-34a has been reported as a predictive marker for docetaxel response in PCa patients [41]. Another recent study has reported specific miRNAs in serum exosomes as biomarkers to predict response to radiotherapy [42].

In addition, PCa-derived EVs have been reported to play a role in metastatic bone remodeling including favoring the differentiation of osteoblasts, impairing the differentiation of osteoclasts, and driving osteomimicry of PCa cells in the bone microenvironment [43,44,45,46,47,48]. Intriguingly, integrins that are found in the exosomal cargo can be instrumental in organ-specific metastasis [49] and the transfer of αVβ6 integrins between PCa cells was shown to promote cell adhesion and migration [50]. In all, these findings highlight the role of exosomal cargo, importantly miRNAs, in driving the metastatic process. It is also important to note that altered miRNA levels within cancer cells have implications for regulating microenvironment behavior, a function that extends beyond transcriptional regulation confined to tumor cells. This is particularly important for the establishment of the pre-metastatic niche in the bone, wherein exosome signaling is a key means of communication amongst normal niche members.

### 1.3. Immune Involvement/Evasion (Coup d’état)

Many of the major innate immune cells and their members, particularly macrophages, dendritic cells, mast cells, and natural killer (NK) cells have been correlated with tumor progression, and patient outcome. In addition, studies using animal models have revealed that therapies targeting these cells have the potential to improve clinical responses in both the traditional cytotoxic and the newer immune-therapeutic modalities. Under normal circumstances, these immune cells function to protect the host from foreign intrusions. In the case of cancer and metastasis in particular, the conversion of these once vigilant defenders to potent co-conspirators of metastatic progression can be equated to instigation of a coup d’état against the host immune responses (Figure 1).

Most, but not all studies of PCa using pathological specimens, have found a correlation of tumor-associated macrophages with worse prognosis [51,52]. In addition, these investigations reported the association of the M2 macrophage phenotype in higher grade tumors [53,54]. Intriguingly, it is thought that M2 macrophages drive other elements in the stroma, particularly in mediating the conversion of quiescent fibroblasts to carcinoma associated fibroblasts (CAFs) which work in alliance with tumor cells to promote PCa progression [55]. In murine models of PCa, macrophage targeting utilizing CSF1 inhibitors to block macrophage migration improved radiation therapy efficacy and restored androgen blockage sensitivity [56,57,58]. In addition, macrophages can play a role in the metastatic progression of PCa as demonstrated by a recent study that showed increased cytokines associated with circulating macrophages in docetaxel-treated patients [59]. Macrophages therefore appear to play a supportive role in PCa progression, and macrophage-targeted therapeutic strategies have shown early signs of efficacy in mouse models thereby highlighting the potential for trials in humans [57].

Natural Killer (NK) cells are innate lymphocytes that rapidly respond to viral infections and other pathogenic insults. NK cells have been demonstrated to mediate cytotoxic activity against tumor cells, and through the secretion of key cytokines and chemokines, regulate the activity of other immune cells. Murine models have revealed that altering the activity or numbers of NK cells is associated with PCa progression and metastasis [60]. This was verified in PCa patients where increased CD56^+^ NK cells following androgen deprivation therapy (ADT) was associated with a better prognosis, whereas low numbers of CD56^+^ NK cells were predictive for seminal vesicle invasion [61]. Other reports have corroborated these findings wherein higher NK cell activation and cytotoxicity correlated with delays in development of castration resistance as well as improved overall survival in metastatic PCa patients [62]. Thus, akin to many cancers, emerging data suggests a crucial role of NK cells in ameliorating the progression of PCa, and in improving therapy response.

Other innate immune cells that contribute to PCa progression include neutrophils and mast cells. Mast cells, inflammatory tissue resident cells that protect against parasites and bacteria through their secretion of immuno-regulatory cytokines, have been reported in PCa and increased numbers of tumor-associated mast cells likely inhibit tumor progression and are predictive of a better prognosis [63,64]. Conversely, mast cells also have been reported to be key mediators of early PCa development as they produce pro-angiogenic and pro-tumorigenic matrix metalloproteinase MMP9 [65]. Therefore, the role of mast cells in PCa progression remains controversial. Along similar lines, neutrophils are also an ambivalent player in disease progression. Although neutrophils, similar to mast cells, appear to correlate with decreased incidence of metastatic PCa as well as response to therapy [66], neutrophils can also secrete MMP9—which has been shown to mediate PCa progression [67]. In all, while macrophages, mast cells, and neutrophils largely exhibit pro-tumorigenic properties, NK cells on the other hand appear to be anti-tumorigenic [68,69,70,71]. However, the contribution of these innate immune cells in PCa progression and metastasis needs to be further evaluated since the precise effects of these cell types can vary depending on the stage and grade of the disease.

T cells and B cells are lymphocytes that make up the adaptive immune system. Most immune responses lead to the activation of these lymphocytes, resulting in the production of cytotoxic cells, cytokines, and antibodies. In PCa, evidence of activation of the adaptive immune system is demonstrated by studies of T cell specificity against prostatic acid phosphatase and prostate-specific antigen—indicating existing memory T cell responses [72]. In accord with their diverse phenotypes, T and B cells have been documented to play complex and multi-faceted roles in cancer—both as promoters of tumor progression, as well as key components of the anti-tumor immune responses [73]. Analysis of prostate tumor infiltrate has revealed that increased numbers of T cells or B cells is indicative of worse prognosis—linked to capsular or perineural invasion [74]. Particularly, these reports showed that increased numbers of CD4^+^ or CD8^+^ T cells correlated to poor survival in PCa patients [74,75,76]. Interestingly, a significant population of these infiltrating T cells are CD4^+^CD25^+^FoxP3^+^ regulatory T cells [77,78]. In addition, these regulatory T cells are elevated in the circulation of PCa patients [79,80], and their presence, while being indicative of poor survival, also predicts adverse responses to immunotherapies including anti-CTLA4 blockade and vaccines [81]. A potential explanation for this is that regulatory T cell homeostasis, essential for immune tolerance, can be disrupted by these therapies, thereby leading to autoimmune events [81]. In sum, infiltration of T cells in PCa in general appears to be predictive of a worse prognosis, likely because a majority of these T cells may have regulatory activity.

B cells have been reported to mediate cancer progression in murine models and regulate therapy responses via their influence on macrophages [82]. B cells are highly enriched in PCa tissue, and their presence positively correlates with higher grade and enhanced risk of recurrence [83], although previous histological analyses suggest that B cell infiltration in lymph nodes may be more indicative as compared with intra-tumoral B cells [84]. In a recent seminal study using PCa mouse models, it was demonstrated that B cells that were recruited by the chemokine CXCL13 into prostate tumors exacerbated castration resistance by producing lymphotoxin, which in turn activates IkB kinase alpha (IKKα)-BM1 mediated prostate regeneration by prostate progenitor cells. [85]. In addition, recent studies using murine models have revealed that B cells that are immunosuppressive in nature could drive resistance to chemotherapy via the production of the immunosuppressive cytokines interleukin-10 and programmed death-ligand (PD-L1) [86]. Therefore, there is emerging evidence in murine models to suggest that B cells may contribute to tumor progression as well as therapy resistance; however, further studies in mouse models and patients are necessary to delineate which B cell subsets are playing roles in progression.

### 1.4. Dormancy at the Secondary Site (Staying Underground and Waiting for the Opportune Moment to Strike)

Although the concept of tumor dormancy was posited nearly a century ago [87], seminal findings within the last 25 years have resulted in an increasing recognition of the potential to utilize this clinical phenomenon as a means to target tumor cells during dormancy [88,89,90]. Technological advances made it possible to detect disseminated tumor cells (DTCs) [88], and the presence of these DTCs in patients with no detectable metastatic disease [88,89,90] suggested that tumor cells spread far earlier than was previously thought, as was corroborated by animal models [91,92]. The fact that DTCs could be used as a tool for prognostic significance [89,90,93,94,95,96,97,98,99] indicated that these DTCs could be the potential origin for future metastatic outgrowths. In addition, the subsequent detection of DTCs in cancer patients sometimes even decades following successful therapy of their primary tumor [100,101,102] suggested that a fraction of DTCs escape treatment and subsist in spite of no clinical evidence of disease. The finding that DTCs can reside at the level of single cells [103,104], and that akin to stem cells, their behavior is controlled by the microenvironment [105], has given rise to the possibility that we can therapeutically target dormant DTCs by altering their niche. In a combat, strategic waiting can be used to maximize the exploitation of weaknesses, or to permit the passage of time to facilitate a lapse by opposing forces. The strategy of DTCs, which entails a prolonged period of dormancy aimed at escaping therapeutic intervention, followed by initiation of metastatic outgrowth, is analogous to waiting in ambush for an opportune time to strike in combat (Figure 1 and Figure 2).

Evidence suggests that similar to HSCs, colonizing tumor cells locate to specialized microenvironments or niches that support cell survival and sustain long-term dormancy [106]. Although the molecular mechanisms that mediate dormancy are poorly defined, it is thought that the pathways that regulate dormancy may be discrete from those that mediate colonization. For example, PCa cells express annexin II receptor which, by binding to annexin II on bone cells, regulates tumor growth [19,106,107]. This interaction in PCa regulates the expression of growth arrest specific 6 (GAS6) receptors—receptor tyrosine kinases AXL, TYRO3 (also called SKY), and MER (also known as MERTK) [107] (Figure 2). It is thought that the expression level of GAS6 and those of the receptors, AXL in particular, may be a key mediator in controlling dormancy, and that high AXL levels is associated with dormancy in human PCa xenograft models [107,108,109]. Intriguingly, hypoxia is known to stabilize AXL [110], and the metaphyseal region of the long bone, a site which is more susceptible to the colonization of cancer, is normoxic, while the diaphyseal region of the bone is more hypoxic [111] and less vulnerable to developing metastasis. Furthermore, endosteal regions are less hypoxic when compared with perivascular regions that are deeper [112]—another facet of bone biology that may be linked to maintaining cells in a dormant condition. The niche is thought to be key in keeping the tumor cells in the state of long term-dormancy in the bone, and this may be a process that involves multiple steps. In the first step, the tumor cells engage with the niche using a variety of receptors and adhesion molecules to bind to the cells of the niche. The second step involves the niche regulating the behavior and phenotype of the colonizing tumor cells in an effort to stabilize and keep them in a dormant state. This is brought about by the induction of a new gene expression in the tumor cells by the cells of the niche. The final and crucial step of releasing the tumor cells from the niche in order to form an overt metastatic lesion likely involves complex regulatory mechanisms that are largely unknown (Figure 2).

It is thought that dormant cancer cells are reactivated due to bone resorption that is brought about by osteoclasts. Interestingly, the vicious cycle model was proposed to explain how the cancer cells are able to grow and form lesions in the bone, and that: a) growth factors within the bone such as bone morphogenic proteins (BMPs) and TGF-β that are released due to osteoclast activity; and b) factors secreted by tumor cells that have colonized the bone, such as parathyroid hormone related protein (PTHrP), that in turn stimulates bone resorption by upregulating receptor activator of nuclear factor kappa-β (RANKL)—viciously feed each other to support tumor growth in the bone microenvironment [113,114]. However, the vicious cycle model does not take into account how reactivation of dormant cells due to bone resorptive activity contributes to the tumor growth, or the initial events that establish the interdependence between the tumor cells and osteoclast activity. A current version of the vicious cycle model (Figure 2)—which is supported by increasing evidence—takes into account the temporal development of the tumor and proposes that osteoclasts start the process by their resorptive activity and thereby remodel the bone microenvironment to release the dormant tumor cells. This is now followed by a vicious cycle between the bone microenvironment and the tumor cells leading to the establishment of the metastatic lesion. Studies that support the refined vicious cycle model demonstrated that mice treated with G-CSF or soluble RANKL (sRANKL) activated bone resorption and resulted in the mobilization of HSCs from the bone niche [115]. Intriguingly, PCa cells home to the endosteal niche by competing with the HSCs treatment with G-CSF mobilizes these cells from the bone niche [19]. Further, CXCL12 interaction with its receptor CXCR4, which is key for the homing of the tumor cells to the bone niche, can be disrupted through cleavage of CXCL12 by cathepsin K—the major osteoclast-produced resorptive proteinase [115]. These reports reinforce the idea that dormant cancer cells can be released from the bone niche by osteoclast resorptive activity, which in turn leads to the establishment of overt metastatic lesions.

The currently used therapeutic approaches to treat PCa may have clinical implications in the context of dormancy and reactivation of cancer cells. For example, it is thought that ADT—the predominant clinical therapeutic option used in PCa patients—is linked to bone loss that is mediated by osteoclasts [116]. This bone loss can be inhibited by the administration of bisphosphonates [117,118,119,120] or the anti-RANKL antibody Denosumab [121,122]. It is posited that ADT may also have unintended and additional consequences that include dormant tumor cell reactivation through the stimulation of resorptive activity by osteoclasts [123]. Additionally, the observation that dormancy can be controlled by bone cells leads to novel therapeutic avenues to treat PCa. For example, inhibiting bone resorption using Denosumab in castrate-resistant PCa patients led to increased bone-metastasis-free survival and an associated delay in the time to first bone metastasis [124]. Furthermore, and of note, recent clinical studies have suggested that treatment with the bisphosphonate, zoledronic acid, initiated at the time of ADT therapy administration, delayed the time of prostate-specific antigen failure [125,126], which was not observed if zoledronic acid treatment was delayed [127]—underscoring the notion that early intervention against bone loss is inhibitory to osteoclast mediated reactivation of dormant PCa cells.

### 1.5. Reprogramming (Hacking the Information Systems)

Many cancers, including PCa, have evolved mechanisms to activate diverse pathways, especially under the selective pressure of targeted therapies. Wide-scale reprogramming is often utilized as a means to circumvent the odds and achieve the goal of tumor progression and metastasis. This reprogramming could be compared to hacking the information systems or re-wiring the information systems to inflict maximum damage (Figure 1). The numerous ways in which cancer cells re-wire host systems include metabolic, stromal, and epigenetic reprogramming. Widespread epigenetic reprogramming in PCa progression, particularly in the context of androgen receptor (AR) signaling, is especially relevant to understand in this context.

Recent studies on chromatin structure and histone post-translational modifications have increasingly appreciated the altered chromatin-binding patterns of AR or other transcription factors as the oncogenic driving forces in PCa progression [128,129,130,131,132]. Supporting this notion, the AR binding sites have been reported to be significantly reprogrammed during PCa progression [130]. A growing consensus points toward epigenetic alterations, including chromatin structure and density changes, as an oncogenic means to alter the cistromes of transcription factors (TFs). Remarkably, the genome-wide set of androgen receptor binding sites (ARBS) was found to be significantly and consistently reprogrammed in PCa in comparison to adjacent normal tissue [133], and ChIP-seq in clinical samples revealed that AR binding to chromatin was enhanced in castrate resistant PCa (CRPC) tissue compared with primary PCa [130]. Further, studies have reported that metastatic CRPC specimens are linked with a higher number of genomic sites showing open chromatin conformation as compared with primary untreated PCa or locally recurrent PCa specimens [132,134]. This suggests extensive chromatin reprogramming during the progression to castrate resistant disease, and inter-patient sample analysis has revealed that the core set of open chromatin regions in the genome, while similar in benign prostatic hyperplasia (BPH) and primary PCa, were varied in CRPC samples [132]. It has been established that Forkhead Box A1 (FOXA1) paves the path to the opening of compact chromatin to facilitate binding of other TFs including AR [135,136,137,138]. Of note, FOXA1 expression has been reported to be associated with tumor progression and poor prognosis in PCa, and interestingly, FOXA1 physically interacts with AR [139,140]. Along similar lines, a recent elegant study points to FOXA1 mediated reprogramming of enhancer regions in metastatic pancreatic cancer, further corroborating the role of transcriptome alteration in metastatic progression [141]. In all, these findings suggest that chromatin relaxation is a key feature of PCa, and that open chromatin configurations are associated with increased transcriptional activity—resulting in reprogramming of the global transcriptional output of cancer cells to drive CRPC progression and metastasis.

In contrast to conventional CRPC adenocarcinomas, the chromatin structure of treatment-induced neuroendocrine CRPCs (t-NEPC)—an emerging therapy-resistant CRPC subtype associated with AR positive or negative states, and a different transcriptomic as well as mutational landscape – has not been extensively studied [142,143,144]. However, t-NEPC is known to harbor distinct molecular changes, and it would be interesting to elucidate if the chromatin remodeling observed in CRPC is maintained or further increased in t-NEPC. Compared to CRPCs, t-NEPCs are reported to harbor RB1 and p53 loss more frequently [145]. Of note, RB1 loss has been associated with cistrome reprogramming of TFs in CRPC [146]. The concomitant loss of p53 and RB1 has been reported to orchestrate the upregulation of chromatin modifying factors such as polycomb repressive complex 2 (PRC2) catalytic subunit enhancer of zeste homolog 2 (EZH2), and SRY (sex determining region Y)—box 2 (SOX2). These facilitators of epigenetic reprogramming are linked to the emergence of t-NEPC. In addition, t-NEPC is associated with N-MYC overexpression. Using in vitro and in vivo models, N-MYC overexpression has been shown to mimic features of NEPC including suppression of AR signaling. The authors suggest that N-MYC may be impacting AR signaling by binding to the enhancer regions of AR in the absence of ligand activated AR [147]. Further, the inhibition of AR signaling by N-MYC seems to be dependent on N-MYC-EZH2 complex which promotes EZH2 activity. Further, both EZH2 and N-MYC are reported to be over-expressed in a majority of NEPC samples [145]. Of note, a study reported that several histone-modifying enzymes, characteristically associated with chromatin remodeling, such as CBX2, EZH2, and polycomb group of proteins with DNA methyltransferase activity (DNMT) were over-expressed in t-NEPC when compared with CRPC adenocarcinomas [148]. Taken together, these studies indicate that the transcriptomes of t-NEPC could intrinsically differ from CRPCs, and these studies point to the reconfiguration of the TF complexes to promote CRPC progression as well as the emergence of the t-NEPC phenotype.

## 2. Conclusions and Future Perspectives

The overwhelming barriers to metastatic colonization of prostate and other cancers are navigated by an increasingly well-understood framework of strategies that share significant and uncanny parallels with military tactics, wherein efficient and deliberate maneuvering, allocation of resources, and timing can be decisive. These strategies generally employ the primary and secondary tumor microenvironments and involve immune cells, fibroblasts, and bone cells as key accomplices, with reciprocal interaction and transcriptional reprogramming playing a key role in each step of cancer progression and metastasis. Intriguingly and as increasingly appreciated, the recurring theme in these strategies involves the utilization of the tumor microenvironment in various stages of the metastatic journey. This includes the ability of the cancer cells to camouflage against the host immune surveillance, the ability to undergo prolonged periods of hibernation or dormancy, and the extremely potent and far-reaching effects of reprogramming the genetic and epigenetic machinery of cancer cells. An evolving understanding of these strategies reiterates the inherent nature of the metastatic process as far from being random or stochastic, and as one that is largely dictated by connivance and deliberate out-maneuvering of the host systems — a theme whose very foundations were laid by Paget’s “seed and soil” hypothesis. The recent years have seen seminal progress in delineating the signaling, molecular, and cellular mechanisms that underpin these strategies used by metastatic cells. The stratification of metastatic progression into a discrete set of strategies, likened to tactics in a military combat setting, will help us in shaping our evolving perspective of the broad principles used by a cancer cell to achieve metastasis with the goal of developing metastasis-specific therapeutic modalities.

## Figures and Tables

**Figure 1 cancers-11-01928-f001:**
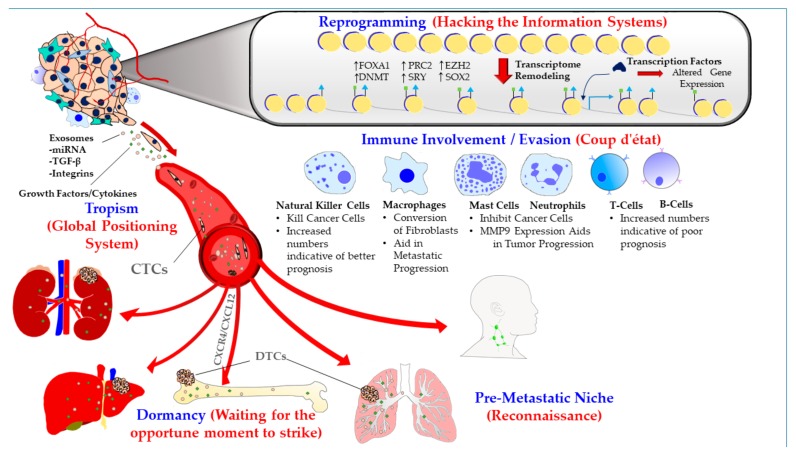
Various strategies devised by prostate cancer cells to overcome the odds of establishing metastasis: Metastasis in prostate cancer (PCa) is driven by several strategies, which include pre-metastatic niche formation, organo-tropism, reprogramming including chromatin remodeling, and immune involvement/evasion, and dormancy at the secondary site. CAFs: cancer associated fibroblasts, miRNA: micro RNA, TGF-β: tumor growth factor β, CTCs: circulating tumor cells, DTCs: disseminated tumor cells, CXCR4: chemokine (C-X-C motif) receptor type 4, CXCL12: (C-X-C motif) chemokine ligand 12, MMP9: matrix metalloproteinase 9, FOXA1: forkhead box A1, PRC2: polycomb repressive complex 2. EZH2: enhancer of Zeste Homolog 2. DNMT: DNA MethylTransferase, SOX2: sex determining region Y- box 2.

**Figure 2 cancers-11-01928-f002:**
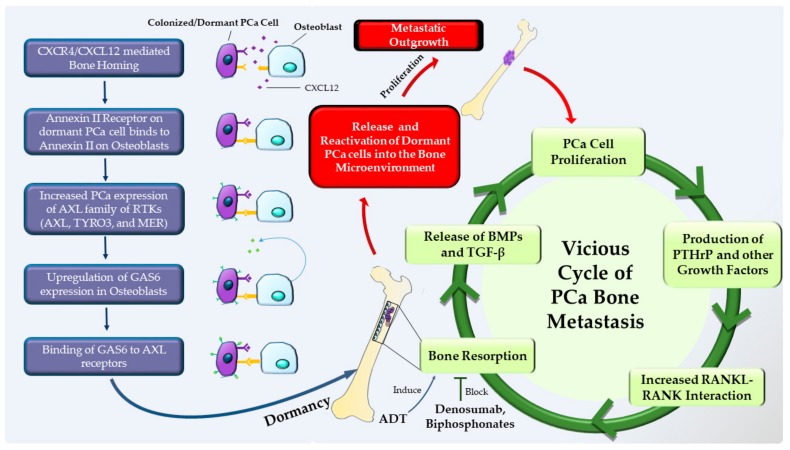
The refined vicious cycle model of prostate cancer bone metastasis that takes into account dormancy and reactivation of prostate cancer cells: release of prostate cancer (PCa) cells from the dormant state is primarily mediated by increased bone resorption, and leads to their reactivation and proliferation, thereby driving metastatic outgrowth in the bone microenvironment. The induction of dormancy in PCa cells is orchestrated by the interaction of receptors expressed on PCa cells [AXL family receptor tyrosine kinases (AXL, TYRO3 and MER), and Annexin II receptor] with the cognate ligands expressed on osteoblasts that include Annexin II and GAS6. The vicious cycle of PCa bone metastasis is mediated by the production of factors by PCa cells that increase bone resorption through enhanced interaction between RANKL expressing osteoblasts and RANK expressing osteoclasts. Release of growth factors such as BMPs and TGF-β from the bone milieu—that is brought about by bone resorption—further fuels the vicious cycle of bone destruction and growth of the metastatic lesions. CXCR4: chemokine (C-X-C motif) receptor type 4, CXCL12: (C-X-C motif) chemokine ligand 12, AXL: AXL receptor tyrosine kinase, TYRO3: TYRO3 protein tyrosine kinase, MER: mer receptor tyrosine kinase, GAS6: growth arrest specific 6.

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
