# Peer review of "How Prostate Cancer Cells Use Strategy Instead of Brute Force to Achieve Metastasis"

_cancers, 2019, doi:10.3390/cancers11121928_

Round 1
Reviewer 1 Report
This is a very well written review regarding metastatic mechanisms in prostate cancer, which draws an analogy to military tactics. This review was a true pleasure to read.
It was a bit distracting that the citations were not in brackets-this made it a little more difficult to ensure all relevant manuscripts were cited.
A table of factors implicated in preparation of a pre-metastatic niche would be helpful, as would a table detailing the specific impact of various tumor cells within the tumor microenvironment in regards to metastatic dissemination.
It seems worthwhile to point out that FOXA1 physically interacts with the AR, and cite the relevant manuscript. PMID: 12750453
A role for FOXA1-driven enhancer reprogramming was recently described in pancreatic cancer, and this manuscript should be cited. PMID: 28757253
Author Response
Thank you for your valuable suggestions. Please see our point-by-point response.

Reviewer 2 Report
In this review article, the authors provide an overview of the metastatic process of prostate cancer making a parallel with the military combat strategies. This is an interesting and original approach. The danger of this approach is finalism, i.e. to make cells appearing as "cleaver entities" that establish strategies to escape defense reactions of the host to eradicate abnormal cells. Although in some instance the wording is borderline (e.g. lines 250 and futher, or "deliberate out-maneuvring" on line 387), the authors are rather successful to avoid this pitfall. The review covers the various steps of metastatic progression applied to the prostate cancer context (metastatic tropism, establishing the pre-metastatic niche, immune system involvement or evasion, dormancy at secondary site, and cell reprogramming). The review is well written, pleasant to read, and very informative.
I only have minor comments or suggestions :
EMT and MET are usually considered as important steps to escape the primary tumor and establish in distant sites, respectively. These process are not mentioned in this review, is there any reason?
Lines 37 and further: Could the authors discuss the fate of the 99.98% of cells that undergo "metastatic failure"? Do they die in the CTC state? Do they die in other organs less adapted to metastatic colonization? Do they die in the host organ where not all tumor cells find the way to establish a metastatic foci? Do they include dormant cells? Please comment based on the state of the art.
Line 21: "Although cancer rates have improved over the past few years". Do the authors mean that less patients have been diagnosed with PCa, which sounds counter-intuitive given the more generalized PSA screening? Or do they mean that PCa patient survival has improved? Please clarify.
Line 134: this reviewer understands that "thereby transferring membrane receptors between cells" is independent of "interactions with ligands expressed on the surface" and rather involves EV/cell membrane fusion. Could the authors clarify this sentence?
Line 181-183: is this conclusive statement supported by previously-cited ref 56-57 or others? Please add appropriate references for clarity.
Lines 219-222: for non-immunologists it is not obvious to link the presence of regulatory T cells to "adverse responses to immunotherapies". While this reviewer understands that immune therapy has yet to demonstrate efficacy in prostate cancer patients, it may be important to slightly develop this part of the review for broad readership.
Line 238: the "three seminal findings" the authors refer to should be clearly identified in the following text.
Line 283: is the revised version of the vicious cycle shown in Fig 2 original to this review article? If so it may be valuable to state it more clearly (e.g.: we here provide a revised version of…").
Line 301: what is the proposed mechanism for reactivation of dormant tumor cells by ADT?
Line 305: please specify that zoledronic acid is a bisphosphonate
Line 323: shift title to next page
Format issues
Reference numbers need to be reformated as superscripts (only ref 1 is correct!). Is the 's grammatically correct on lines 33 (cancer cell's execution), 350 (cell's global transcriptional output), 382 (cell's metastatic journey), 385 (cell's genetic and epigenetic machinery)? All abbreviations should be defined (some are not) but should be defined only once (e.g. HSC is defined 3 times). Text in Figure 1 is quite small, it would be welcome to use bigger font. Fig 2: would it be possible to include on the figure – or make a separate one – the actions, interactions and target of ADT/bisphosphonate/osteoclasts/etc. on the vicious circle to better understand how these treatments can affect the metastatic process and therapeutic outcome (side effects)?Author Response
Thank you for your valuable suggestions. Please see our point-by-point response.
